# Effects of Synthesis Conditions of Na_0.44_MnO_2_ Precursor on the Electrochemical Performance of Reduced Li_2_MnO_3_ Cathode Materials for Lithium-Ion Batteries

**DOI:** 10.3390/nano14010017

**Published:** 2023-12-20

**Authors:** Ya Sun, Jialuo Cheng, Zhiqi Tu, Meihe Chen, Qiaoyang Huang, Chunlei Wang, Juntao Yan

**Affiliations:** 1College of Chemistry and Environmental Engineering, Wuhan Polytechnic University, Wuhan 430023, China; 13995967507@163.com (J.C.); tzq061721@163.com (Z.T.); 15327194236@163.com (M.C.); wangcl@whpu.edu.cn (C.W.); 2School of Textile Science and Engineering, Wuhan Textile University, Wuhan 430200, China; 13986938081@163.com

**Keywords:** Na_0.44_MnO_2_, Li_2_MnO_3_, nanobelts, molten-salt method, rate performance, cathode material

## Abstract

Li_2_MnO_3_ nanobelts have been synthesized via the molten salt method that used the Na_0.44_MnO_2_ nanobelts as both the manganese source and precursor template in LiNO_3_-LiCl eutectic molten salt. The electrochemical properties of Li_2_MnO_3_ reduced via a low-temperature reduction process as cathode materials for lithium-ion batteries have been measured and compared. Particularly investigated in this work are the effects of the synthesis conditions, such as reaction temperature, molten salt contents, and reaction time on the morphology and particle size of the synthesized Na_0.44_MnO_2_ precursor. Through repeated synthesis characterizations of the Na_0.44_MnO_2_ precursor, and comparing the electrochemical properties of the reduced Li_2_MnO_3_ nanobelts, the optimum conditions for the best electrochemical performance of the reduced Li_2_MnO_3_ are determined to be a molten salt reaction temperature of 850 °C and a molten salt amount of 25 g. When charge–discharged at 0.1 C (1 C = 200 mAh g^−1^) with a voltage window between 2.0 and 4.8 V, the reduced Li_2_MnO_3_ synthesized with reaction temperature of Na_0.44_MnO_2_ precursor at 850 °C and molten salt amounts of 25 g exhibits the best rate performance and cycling performance. This work develops a new strategy to prepare manganese-based cathode materials with special morphology.

## 1. Introduction

As the energy crisis and environmental degradation are intensifying, countries around the world have made carbon reduction commitments to achieve carbon neutrality [1,2]. Developing new energy, realizing energy transformation, and building a green and low-carbon energy system are important measures to reduce carbon dioxide emissions and achieve global carbon neutrality. Lithium-ion batteries (LIBs) stand out in the fields of energy storage devices (electric vehicles, hybrid electric vehicles) and intermittent renewable energy storage because of their high energy density, low cost, and stable cycle life [3,4,5,6,7,8]. As a critical member of LIBs, cathode materials play an important role in the electrochemical performance of LIBs [9,10,11]. Among the potential cathodes, Li-rich layered oxide has been under intense investigation because of its low cost, low toxicity, high specific capacity, and high energy density [12,13,14].

As the terminal member of LLO materials, Li_2_MnO_3_ has been an ideal cathode candidate with a theoretical specific capacity of 459 mAh g^−1^, rich Mn resource, and low cost [15,16]. However, different from other Mn-based cathode materials, the valence of Mn in Li_2_MnO_3_ is +4 and cannot be further oxidized, resulting in no electrochemical activity in the common voltage range (2.0~4.4 V) [17,18]. It is usually electrochemically activated at a high cut-off voltage (4.8 V) during the initial charging process, which results in serious capacity and voltage decay [19,20]. Consequently, other strategies have been proposed to activate or enhance the electrochemical properties of Li_2_MnO_3_. One is to partially replace Li, Mn, and O with other elements [21,22,23]. Chen et al. [21,22] explored the influence of the substitution of Nb and Mo on the electrochemical performance of Li_2_MnO_3_. The results showed that the improved electrochemical properties were attributed to the additional electronic compensation provided by Nb^5+^ and Mo^3+^ for Li_2_MnO_3_, and retarded the release of oxygen during discharging process. The spinel phase LiMn_2_O_4_ was introduced into Li_2_MnO_3_ to form a layered-spinel integrated composite by Wu and He et al. [24,25], exhibiting extremely high specific capacity and excellent high-rate performance. Li_2_MnO_3_ in the integrated structure improves the high specific capacity, while LiMn_2_O_4_ provides a three-dimensional lithium-ion transport channel for the composite, thus enhancing the ratability [24,25,26]. Moreover, the chemical thermal reduction method is also utilized to propose the outstanding electrochemical properties of layered lithium-rich cathodes [26,27,28,29].

Particle size reduction or controlled special morphology has been an effective strategy to improve the electrochemical performance of Li-rich materials [30,31,32,33]. Particle size reduction can reduce the diffusion distance of lithium ion, and nanostructured materials have unique structural which can provide a favorable one-dimensional electron pathway [34].

Na_0.44_MnO_2_ with nanoribbons and nanowires has been used as both the Na source and precursor to prepare lithium–manganese oxides with the corresponding morphology [35,36]. The molten salt method can realize the preparation of anisotropic powder materials with specific components at low reaction temperatures or in a short time as a method of synthesizing inorganic compounds [37,38]. The structure of the obtained materials is relatively uniform with no or only weak agglomeration, and nanomaterials with specific morphology can be synthesized by adjusting the reaction conditions. The factors affecting the morphology of the materials prepared via the molten salt method mainly include the type and amount of salt, reaction temperature, and reaction time, etc. [39]. Zhang et al. [40] synthesized Li_2_MnO_3_ nanoribbons via the precursor template Na_0.44_MnO_2_ nanoribbons with the mixture of LiNO_3_-LiCl as molten salt, and studied the magnetic behaviors. Our previous works have proven that Li_2_MnO_3_ nanobelts/nanowires can be successfully prepared using a Na_0.44_MnO_2_ precursor, and the low-temperature reduction products exhibit improved electrochemical performance [27,28,29]. However, the relationships between the synthesis conditions of a Na_0.44_MnO_2_ precursor and the electrochemical performances of reduced Li_2_MnO_3_ could be further discussed.

Here, the Na_0.44_MnO_2_ nanobelt precursor was fabricated via the molten salt method, and then the precursor was lithiated to obtain Li_2_MnO_3_ nanobelts. The influences of reaction temperature and molten salt contents on the crystallinity and morphology of Na_0.44_MnO_2_ precursor, as well as on the electrochemical performance of low-temperature-reduced products were also discussed. In addition, the growth mechanism of Na_0.44_MnO_2_ nanobelt precursor was investigated by adjusting the reaction time.

## 2. Experimental

### 2.1. Material Synthesis

Synthesis of Na_0.44_MnO_2_ nanobelts: Analytical NaCl was used as molten salts without further treatment. Na_2_CO_3_ and MnCO_3_ were fully ground with a certain amount of NaCl in an agate mortar according to the stoichiometric ratio; then, the ground powders were transferred to the crucible. The mixture reacted in a muffle furnace at 800–900 °C for a certain time. After cooling to room temperature, the calcined product was filtered, washed with a large amount of deionized water, and dried at 100 °C.

Synthesis of Li_2_MnO_3_ nanobelts: Analytical LiNO_3_ and LiCl mixture were used as molten salts without further treatment. At the same time, LiNO_3_ was used as the oxidizing reagent. Na_0.44_MnO_2_ precursor was fully ground with 88 mol% LiNO_3_ and 12 mol% LiCl in an agate mortar, then transferred to the crucible. The mixture was sintered at 500 °C for 1 h. After cooling to room temperature, the sintered product was washed with a large amount of deionized water, and dried at 100 °C. In this process, Li_2_MnO_3_ involved the ion exchange between Li^+^ in molten LiNO_3_ and LiCl salts and Na^+^ in Na_0.44_MnO_2_ and the oxidation process by LiNO_3_ according to previous work [40].

Low-temperature reduction of Li_2_MnO_3_ nanobelts: The reduced Li_2_MnO_3_ nanobelts were synthesized according to our previous work [27,28,29]. The prepared Li_2_MnO_3_ nanobelts, stearic acid, and an appropriate amount of absolute ethanol were ground for a few minutes to obtain a uniform solid–liquid rheological material. Then, the obtained rheological material was heated at 340 °C in an Ar atmosphere for 8 h. After washing several times, the product was obtained. The mol ratio of Li_2_MnO_3_ nanobelts and stearic acid was 1:0.04.

### 2.2. Materials Characterization

The crystal structures of the as-prepared samples were conducted on a Bruker D8 ADVANCE (Berlin, Germany) X-ray diffraction via Cu-Kα radiation. The as-prepared Na_0.44_MnO_2_ samples were operated over a 2θ range of 5–60° at rates of 5°/min; Li_2_MnO_3_ and low-temperature reduction of Li_2_MnO_3_ were operated over a 2θ range of 10–80° at rates of 5°/min. The morphology of the as-synthesized samples was characterized using scanning electron microscopy (SEM, FEI, SIRION, Eindhoven, The Netherlands) with a working voltage of 20 kV.

### 2.3. Electrochemical Tests

To fabricate the electrodes, the as-synthesized samples were mixed with super P and polytetrafluoroethylene (PTFE) at a ratio of 80:15:5 onto Al foil. In an Ar-filled glovebox, the batteries were assembled using lithium foil, Celgard 2300 microporous films, 1 M LiPF_6_ in the ethylene carbonate (EC), and dimethyl carbonate (DMC) (1:1 *v*/*v*) as anodes, separators, and electrolytes, respectively. The electrochemical tests were performed on a Neware battery test system (Shenzhen, China) at a voltage window between 2.0 and 4.8 V, ranging from 0.1 C to 5 C.

## 3. Results and Discussion

### 3.1. Effects of Molten Salt Reaction Temperature

In chemical reactions, temperature has a great influence on reaction rates. At higher temperatures, the motion rate of molecules or ions can be accelerated, and the average kinetic energy and the collision frequency of molecules will accelerate in unit time. Therefore, the temperature of the synthesis process plays a vital role in the crystallinity of products [41].

Figure 1 shows the XRD patterns of Na_0.44_MnO_2_ synthesized at 800, 850, and 900 °C (denoted as NMO-800, NMO-850, and NMO-900) via the molten salt method. It can be seen from the figure that the main phase of the samples prepared at different temperatures is the characteristic peak of orthogonal phase Na_4_Mn_9_O_18_ (JCDS. No. 27-0750) [40], and the Na_0.91_MnO_2_ impurity phase appears in the samples. With the increase in reaction temperature, the characteristic peaks of Na_0.44_MnO_2_ are continuously enhanced and the crystallinity becomes stronger.

SEM images of Na_0.44_MnO_2_ precursor synthesized at different temperatures are shown in Figure 2. The as-prepared samples all exhibit the nanobelts’ morphology with lengths of one to several tens of micros, and with a width that varies with different temperatures. When the reaction temperature is 800 °C, the average width of Na_0.44_MnO_2_ nanobelts is about 200 nm, and a small number of nanosheets appear in the sample, while the average width is about 240 nm with a reaction temperature of 850 °C. When the temperature rises to 900 °C, the average width of nanobelts is about 250 nm with particles agglomerated. As the reaction temperature increased from 800 to 900 °C, the average width of the samples increased, which is mainly because the increased reaction temperature promotes the growth of the crystal. Additionally, the increase in temperature may lead to an increase in heating time and cooling, which is equivalent to prolonging the reaction time and accelerating the growth of the sample. In general, the suitable temperature of Na_0.44_MnO_2_ is 850 °C because the Na_0.44_MnO_2_ nanobelts synthesized at that temperature are the most uniform and have no agglomeration.

From Figure 3, it is obvious that both Li_2_MnO_3_ (marked as LMO-800, LMO-850, and LMO-900) and the reduced R-Li_2_MnO_3_ (denoted as R-LMO-800, R-LMO-850, and R-LMO-900) show sharp peaks and flat bases, indicating that the synthesized samples have high crystallinity. All XRD diffraction peaks, which can be seen from Figure 3a, are indexed to the monoclinic phase Li_2_MnO_3_ with a space group of C2/m [42]. The feature of the XRD pattern is the superstructure in a range of 20~25°, which is related to the ordered arrangement of Li/Mn in the transition metal layer. After reduction, the diffraction peaks of these superstructures still exist in the samples with weak intensity, suggesting that the low-temperature reduction process has a certain effect on the superstructure. It is worth noting that the intensity of diffraction peaks also increases with increasing temperature. In addition, several new peaks appear in Figure 3b, such as at 15.5° and 61.3°, which were ascribed to the (010) and (221) planes of orthorhombic LiMnO_2_.

SEM images of Li_2_MnO_3_ nanobelts and the reduced R-Li_2_MnO_3_ nanobelts are displayed in Figure 4. It can be clearly seen that Li_2_MnO_3_ still maintains the morphology of the Na_0.44_MnO_2_ nanobelt precursor, with a length of one to several microns. The average width of LMO-800 is about 150 nm, while the LMO-850 is 190 nm and LMO-900 is 220 nm. Among the three materials, LMO-850 has the most uniform particle size and the smallest width. After the lithiation of Na_0.44_MnO_2_, the length and width of the material decrease, and the surface of material becomes rough, which is mainly due to the transition from the tunnel structure of Na_0.44_MnO_2_ to the layered structure of Li_2_MnO_3_. Figure 4d–f shows a series of samples of Li_2_MnO_3_ after low-temperature reduction. After reduction, the samples are still nanobelts, demonstrating that the low-temperature reduction process has no effect on the morphology. However, the length of the material is shorter than that before reduction, and some layered structures are broken. The average width of R-LMO-800 is 170 nm, the average width of R-LMO-850 is 180 nm, and the average width of R-LMO-900 is 190 nm. The particle size of cathodes affects electrochemical properties; therefore, the three samples will exhibit different electrochemical performance.

To investigate the influence of the synthesis temperature of the precursor on the final products, galvanostatic charging–discharging tests were performed at room temperature.

Figure 5a,b exhibits the initial classical charging–discharge curves and the corresponding cycle performance of three reduced samples under the voltage window of 2.0~4.8 V at 0.1 C (1 C = 200 mAh g^−1^). From Figure 5a,b, it can be seen that the initial discharge-specific capacities of R-LMO-800, R-LMO-850, and R-LNO-900 can reach 154.5, 162.4, and 185.3 mAh g^−1^, respectively. After 100 cycles, the discharge-specific capacities of the three samples are 140.3, 158.3, and 147.4 mAh g^−1^, with corresponding capacity retention rates of 90.8%, 97.4%, and 79.6%, respectively. Additionally, the curves of the three reduced samples are quite different from pristine Li_2_MnO_3_, which has been confirmed in our previous works [27,28,29]. The rate performance of three samples at rates of 0.1 C, 0.5 C, 1 C, 2 C, and 5 C with each rate cycling for 20 cycles is shown in Figure 5c. It is clearly shown that the ratability of R-LMO-850 is better than that of the other two samples at various rates. However, at high rates of 2 C and 5 C, the performance of R-LMO-800 and R-LMO-900 is quite similar. The discharge-specific capacities of RLMO-850 at 0.5 C, 1 C, 2 C, and 5C are 136.4, 121.1, 106.6, and 82.5 mAh g^−1^, respectively.

Obviously, the cyclability and ratability of R-LMO-850 are the best when the reaction temperature of the Na_0.44_MnO_2_ precursor is 850 °C. Nevertheless, the initial discharge-specific capacity of R-LMO-850 is slightly lower than that of R-LMO-900. From the XRD patterns and SEM images of the low-temperature-reduced samples, it can be seen that the crystallinities are becoming higher with the increase in reaction temperature. Some studies reported that higher crystallinity could promote the transport of lithium ion since higher crystallinity indicated fewer defects and stacking faults [43]. Moreover, smaller particles can reduce the polarization of electrodes during the charging–discharging process, and then improve the specific capacity. The uniformly disturbed particles can effectively utilize each particle, increase the discharge depth, and finally enhance the electrochemical performance [44,45]. R-LMO-850 nanobelts show smaller sizes than those of R-LMO-900. With the uniformed particle and small particle size, R-LMO-850 effectively utilizes each particle, and shortens the diffusion distance of lithium ions during the charge and discharge process, along with the best performance. Meanwhile, R-LMO-850 exhibits better crystallinity than R-LMO-800, resulting in better electrochemical capacities than R-LMO-800.

### 3.2. Effects of Molten Salt Contents

Figure 6 displays the XRD patterns of the Na_0.44_MnO_2_ precursor synthesized with different molten salt contents (denoted as NMO-20, NMO-25, and NMO-30). All samples correspond to the orthorhombic phase of Na_4_Mn_9_O_18_ with a space group of *Pbam*. The intensities of the diffraction peaks have no obvious differences with changes in the amounts of molten salt, which indicates that the molten salt contents in the reaction process make no difference in the crystallinity of materials. At the same time, the reflection peaks of all the samples are obvious and sharp with good crystallinity.

SEM images and the corresponding particle size distribution of the Na_0.44_MnO_2_ precursor synthesized with different molten salt content are shown in Figure 7. It is obvious that all the samples are nanobelts with a length of one to several tens of microns. The change in molten salt content has little effect on the length of the samples; however, it is quite different for the width. According to figures of particle size distribution, the average widths of NMO-20, NMO-25, and NMO-30 are 240, 207, and 280 nm, respectively, with relatively wide particle size distribution. Moreover, NMO-25 exhibits the smallest particle size.

A series of Li_2_MnO_3_ samples were obtained via lithiation of Na_0.44_MnO_2_ precursors synthesized using different amounts of molten salt, which were labeled LMO-20, LMO-25, and LMO-30, respectively. Then, the lithiated samples were reduced at low temperatures to obtain the final products, marked as R-LMO-20, R-LMO-25 and R-LMO-30, respectively. The corresponding XRD patterns are shown in Figure 8. It can be seen that the samples after lithiation and the final products after low-temperature reduction can be assigned to monoclinic Li_2_MnO_3_ (PDF. 86-1634) with a space group of C2/m. The reflection peaks of all the samples are sharp, suggesting that the crystallinity of the samples is good. Additionally, there is little effect on the crystallinity of all the samples with different amounts of molten salt. After low-temperature reduction, the diffraction peaks at 15.5° and 61.3°, which can be assigned to the (010) and (221) planes of orthorhombic LiMnO_2,_ appear in the pattern.

Figure 9a–c exhibits SEM images of Li_2_MnO_3_ synthesized via Na_0.44_MnO_2_ with different molten salt contents. As can be seen, the lithiation process has no influence on the morphology of materials. Li_2_MnO_3_ still retains the morphology of nanobelts with decreased length. The average particle sizes of LMO-20, LMO-25, and LMO-30 are 250, 210, and 280 nm, respectively. LMO-25 has the smallest particle size and a relatively uniform distribution. After low-temperature reduction, the samples also maintain the nanobelts’ morphology, indicating undamaged morphology during the reduction process. R-LMO-20, R-LMO-25, and R-LMO-30 show particle sizes of 240, 220, and 260 nm, respectively.

Figure 10a,b presents the initial charging–discharging curves of R-LMO-5, R-LMO-20, R-LMO-25, and R-LMO-30 and the corresponding 30 cycles’ performance at 0.1 C. The specific discharge capacities of R-LMO-20, R-LMO-25, and R-LMO-30 in the first cycle are 132.6, 176.8, and 163.8, respectively. After cycling for 100 cycles, the specific discharge capacities of all the samples undergo a process of rising and then falling. In the 100th cycle, the specific discharge capacities are 90.7, 154.2, and 145.3 mAh g^−1^, respectively. The rate performance of all the samples cycled for 10 cycles at rates varying from 0.1 C to 5 C is shown in Figure 10c. Note that R-LMO-25 possesses the best performance at various rates, R-LMO-20 is the second, and R-LMO-20 is the worst. It can be concluded that when the amount of molten salt in the reaction system is 25 g, the specific capacity, cycle performance, and ratability of the final product are the best because R-LMO-25 has the smallest particle size and the most relatively uniform particles among the three samples.

### 3.3. Growth Mechanism of Na_0.44_MnO_2_ Nanobelts

In order to study the growth mechanism of Na_0.44_MnO_2_ nanobelts, the reaction temperature was controlled at 850 °C, and the molten salt content was 5 g; the XRD and SEM images of different reaction times are shown in Figure 11. At the initial stage of reaction, that is, when the reaction time is 60 s, it can be seen from the XRD pattern that the main phase of the product is Na_0.91_MnO_2_, and there are also fewer characteristic peaks of Na_0.44_MnO_2_. It is reported that in the process of synthesizing tunnel-structured sodium manganese compounds, Na_0.91_Mn0_2_ usually appears as the intermediate phase or precursor with a layered structure [46,47]. From the corresponding SEM images, it can be found that the products are mainly nanosheets and there are very few nanobelts, indicating that the nanosheets and nanobelts may be Na_0.91_MnO_2_ and Na_0.44_MnO_2_, respectively. The intensities of Na_0.91_MnO_2_ peaks weaken, while those of Na_0.44_MnO_2_ peaks enhance upon increasing the reaction time to 10 min. Most of the products are nanosheets, with a small number of nanobelts. With the reaction time increasing to 30 min and 60 min, the characteristic peak of Na_0.44_MnO_2_ in the product is continuously enhanced with the morphology of nanobelts. At last, when the reaction time is 5 h, the product is completely Na_0.44_MnO_2_.

Recently, there have been many reports on the growth mechanism of special morphologies. Li et al. [48] studied the hydrothermal preparation of Na_0.5_MnO_2_ nanowires, in which Na-Birnessite was the intermediate phase. In this work, nanowires were formed through the fracture of nanosheets. Wang et al. [47] investigated the growth mechanism of MnO_2_ nanowires by combining the results of XRD and TEM, indicating that nanowires were formed by crimping nanosheets. However, in the process of synthesizing Na_0.44_MnO_2_ nanobelts, no signs of the breakage of nanosheets were found. Therefore, it could be speculated that Na_0.44_MnO_2_ nanobelts were formed by crimping Na_0.91_MnO_2_ nanosheets. Moreover, Na_0.44_MnO_2_ may be grown according to the Ostwald ripening mechanism, that is, a crystal growth model in which large particles grow and small particles dissolve [49,50]. In the process of synthesizing Na_0.44_MnO_2_ nanobelts, the possible reactions are as follows:(1)$$\mathrm{fast}:\;{MnCO}_{3}+{Na}_{2}{CO}_{3}+O_{2}\to {Na}_{0.44}MnO_{2}+{CO}_{2}$$
(2)$$\mathrm{slow}:\;{MnCO}_{3}+NaCl+O_{2}\to {Na}_{0.44}MnO_{2}+{Cl}_{2}$$

The first step is very quick as it can usually be completed in a few minutes, while the second step is completed in a few hours. At the beginning of the reaction, when the amount of MnCO_3_ is sufficient, the first step of the reaction occurs, which explains that the Na_0.44_MnO_2_ phase appears in the system when the reaction time is 60 s. As the time increases to 60 min, small particles grow into nanosheets, some nanobelts develop, and some nanosheets curl into nanobelts. When the reaction reaches 1 h, the Na_2_CO_3_ reaction is completed, and the second step of the reaction occurs. At this time, the Na_0.91_MnO_2_ phase gradually disappears, and the reaction process is slow and lasts for a long time.

As shown in Figure 12, the formation process of a Na_0.44_MnO_2_ nanobelt can be summarized as follows: Firstly, the Na_0.91_MnO_2_ nanosheet with a layered structure is formed rapidly through the reaction. This process may be that sodium ion is dissolved in MnCO_3_ particles, structural transformation occurs at the same time, and there is a small amount of nanobelts in the reaction; then, the nanosheet curls to form a nanostrip, together with the transition from the layered structure to the tunnel structure at the same time, and the nano strip grows according to the Ostwald ripening mechanism.

## 4. Conclusions

In summary, Li_2_MnO_3_ nanobelts were prepared via the molten salt method with Na_0.44_MnO_2_ nanobelts as precursors at different synthesis conditions, characterized by XRD and SEM, and the reduced Li_2_MnO_3_ samples were tested as cathodes for lithium-ion batteries. When the reaction temperature was 850 °C, Na_0.44_MnO_2_ precursor was uniformly distributed with a smaller particle size of 240 nm. The effect of molten salt contents on the crystallinity was not significant; when the molten salt content was adjusted to 25 g, the Na_0.44_MnO_2_ precursor showed the smallest particle size of 204 nm. By comparing the electrochemical performances of reduced Li_2_MnO_3_, the cyclability and ratability of R-LMO-850 are the best when the reaction temperature of the Na_0.44_MnO_2_ precursor is 850 °C; when the amount of molten salt in the reaction system is 25 g, the specific capacity, cycle performance, and ratability of the final product are the best. The synthesis temperature of 850 °C and the molten salt amount of 25 g were considered the optimum synthesis conditions for the Na_0.44_MnO_2_ precursor. The growth mechanism of nanobelts was investigated by controlling the reaction time of the Na_0.44_MnO_2_ nanobelt precursor. The results indicate that Na_0.44_MnO_2_ nanobelts are curled from Na_0.91_MnO_2_ nanosheets. The growth mechanism has guiding significance for the synthesis of materials with special morphologies using the molten salt method. These findings are expected to provide a feasible approach for preparing special morphology cathodes for LIBs.

## Figures and Tables

**Figure 1 nanomaterials-14-00017-f001:**
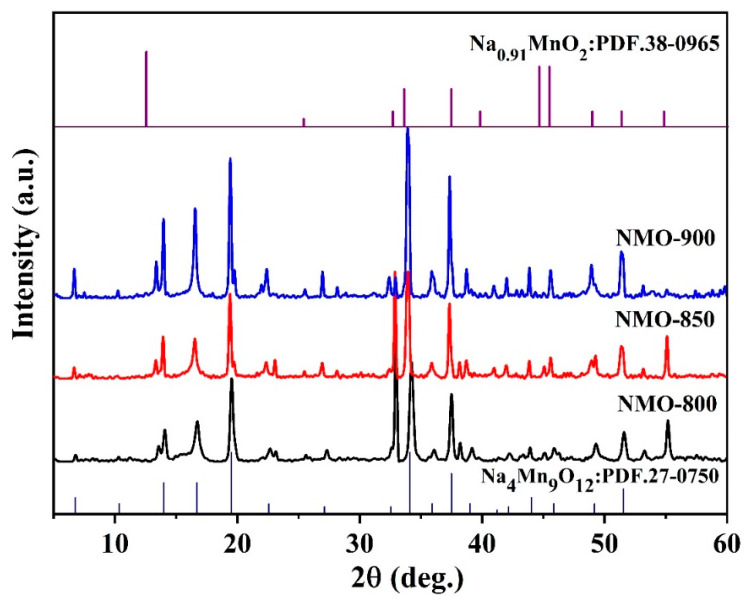
The XRD patterns of NMO-800, NMO-850 and NMO-900.

**Figure 2 nanomaterials-14-00017-f002:**
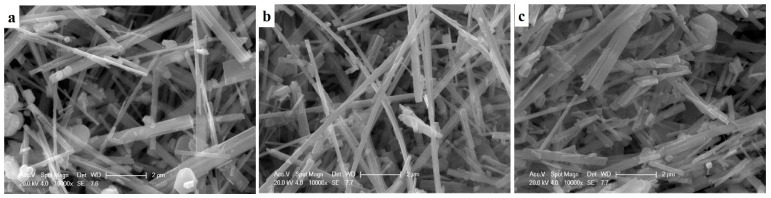
SEM images of NMO-800 (**a**), NMO-850 (**b**) and NMO-900 (**c**).

**Figure 3 nanomaterials-14-00017-f003:**
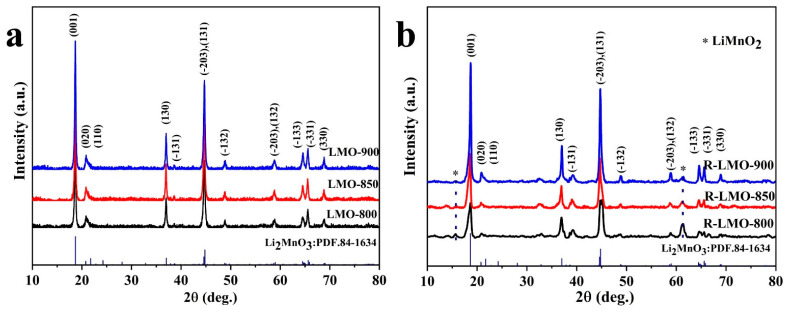
XRD patterns of Li_2_MnO_3_ nanobelts (**a**) and the reduced R-Li_2_MnO_3_ nanobelts at different temperatures (**b**).

**Figure 4 nanomaterials-14-00017-f004:**
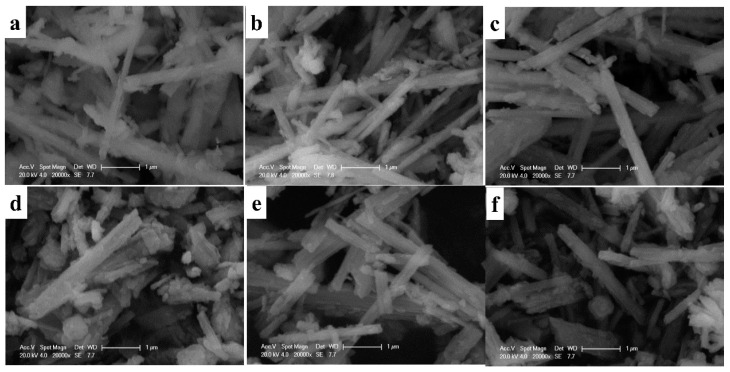
SEM images of Li_2_MnO_3_ nanobelts, (**a**) LMO-800, (**b**) LMO-850 and (**c**) LMO-900, and the reduced R-Li_2_MnO_3_ nanobelts, (**d**) R-LMO-800, (**e**) R-LMO-850 and (**f**) R-LMO-900.

**Figure 5 nanomaterials-14-00017-f005:**
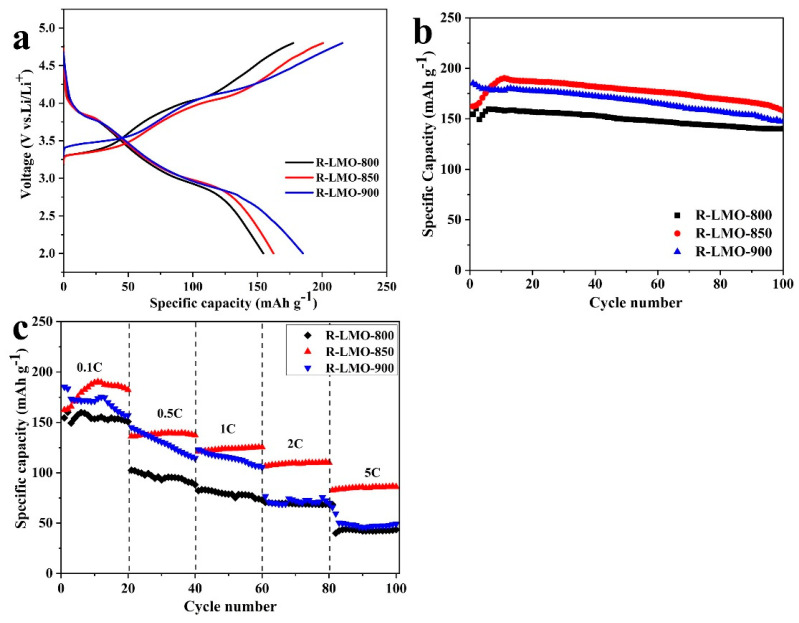
The classical initial charging–discharging curves (**a**), the corresponding cycle performance (**b**) of R-LMO-800, R-LMO-850 and R-LMO-900 at 0.1 C rate, and the rate performance at different rates (**c**).

**Figure 6 nanomaterials-14-00017-f006:**
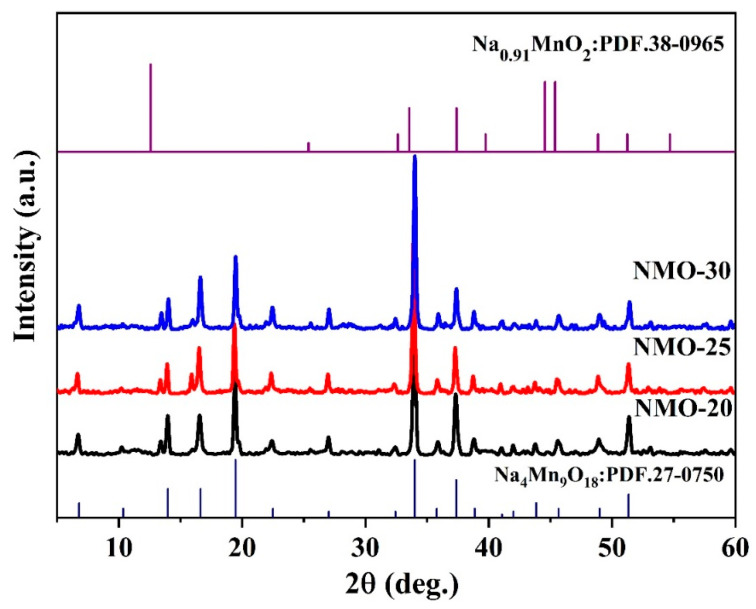
XRD patterns of Na_0.44_MnO_2_ precursor synthesized with different molten salt content.

**Figure 7 nanomaterials-14-00017-f007:**
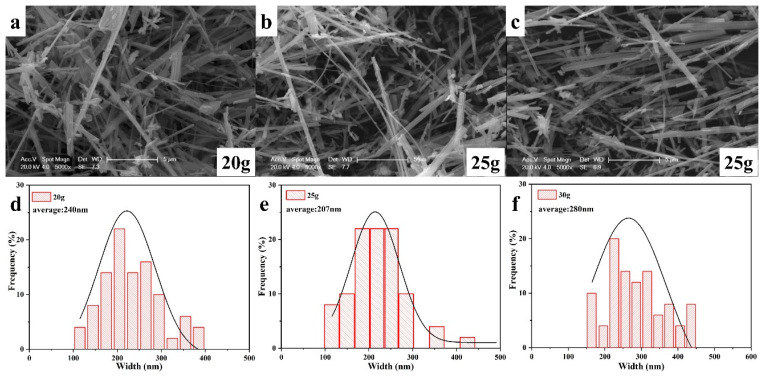
SEM images and size distributions of NMO-20 (**a**,**d**), NMO-25 (**b**,**e**), and NMO-30 (**c**,**f**).

**Figure 8 nanomaterials-14-00017-f008:**
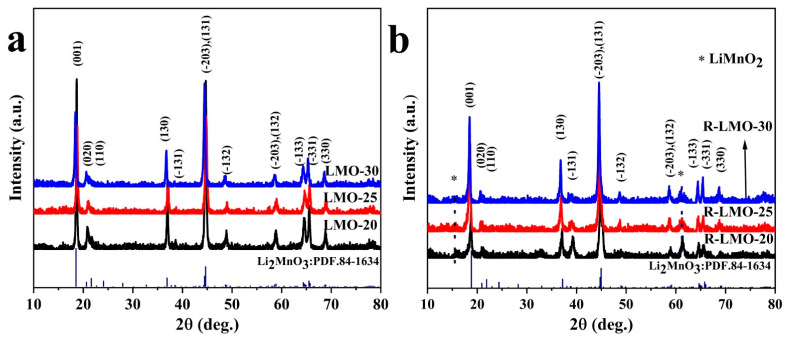
XRD patterns of Li_2_MnO_3_ nanobelts (**a**) and the reduced R-Li_2_MnO_3_ nanobelts with different molten salt content (**b**).

**Figure 9 nanomaterials-14-00017-f009:**
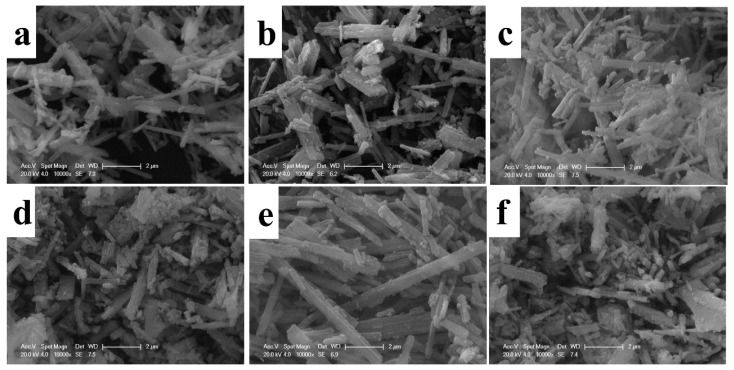
SEM images of Li_2_MnO_3_ synthesized via Na_0.44_MnO_2_ with different molten salt contents: (**a**) LMO-20, (**b**) LMO-25, (**c**) LMO-30 and reduced R-Li_2_MnO_3_: (**d**) R-LMO-20, (**e**) R-LMO-25, and (**f**) R-LMO-30.

**Figure 10 nanomaterials-14-00017-f010:**
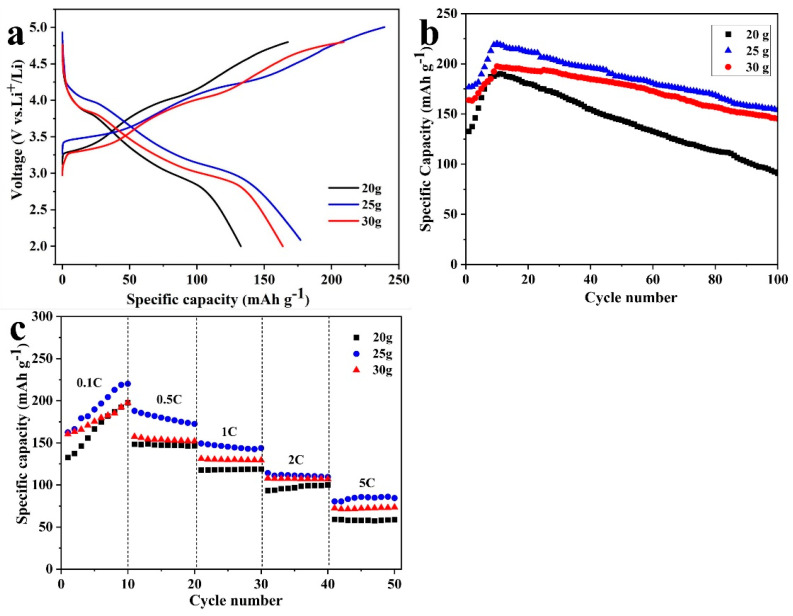
The classical initial charging–discharging curves (**a**), the corresponding cycle performance (**b**) of R-LMO-20, R-LMO-25 and R-LMO-30 at 0.1 C rate, and the rate performance at different rates (**c**).

**Figure 11 nanomaterials-14-00017-f011:**
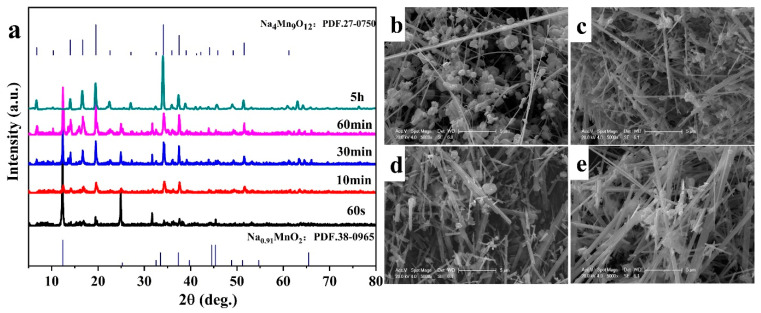
XRD patterns of Na_0.44_MnO_2_ at different reaction time (**a**) and SEM images of reaction for 60 s (**b**), 10 min (**c**), 30 min (**d**), and 60 min (**e**).

**Figure 12 nanomaterials-14-00017-f012:**
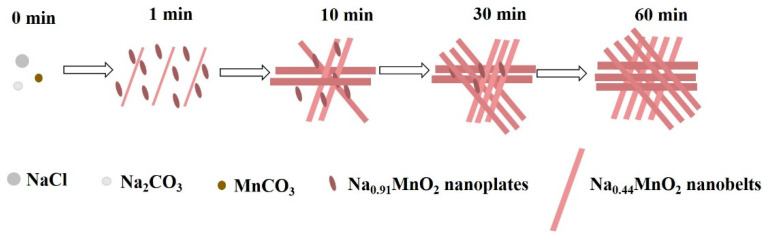
Proposed growth mechanism of Na_0.44_MnO_2_ nanobelts.

## Data Availability

Data are contained within the article.

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
