# Peer review of "Effects of Synthesis Conditions of Na0.44MnO2 Precursor on the Electrochemical Performance of Reduced Li2MnO3 Cathode Materials for Lithium-Ion Batteries"

_nanomaterials, 2023, doi:10.3390/nano14010017_

Round 1

Reviewer 1 Report

Comments and Suggestions for Authors

The authors are not sucessful in delivering a new story on the basis of their work. This should be dug in very much further. For example:

1) Why 25g? Does this depend on the size of the reactor? This should be really clarified.

2) The presentation of XRD results is poor. For example: fig 8. Baseline correction should be applied. Peaks should be identified.

3) Fig 5 and 10: it is better to show how the Li2MnO3 has been activated during first 10 cycles. This is because the initial discharge at 4.8V to about 4V is important. It will confirm that the Li2MnO3 is really reactive or you simply has a capacitor effect.

Comments on the Quality of English Language

It is fairly OK.

Author Response

  1. Summary

Thank you very much for taking the time to review this manuscript. Those comments are all valuable and very helpful for revising and improving our paper, as well as important guides for our research. We have studied the comments carefully and have made corrections. Revised portions are marked in red on the paper.

2. Questions for General Evaluation

Reviewer’s Evaluation

Response and Revisions

Does the introduction provide sufficient background and include all relevant references?

Can be improved

We have revised introduction, for the detailed revisions, please refer to the document named “Revised manuscript with marked changes”.

Are all the cited references relevant to the research?

Can be improved

The references relevant to the research have been added in the manuscript, for the detailed revisions, please refer to the document named “Revised manuscript with marked changes”.

Is the research design appropriate?

Can be improved

Very sorry about that the experiment cannot be further improved due to time limited.

Are the methods adequately described?

Can be improved

We have modified the experimental section, for the detailed revisions, please refer to the document named “Revised manuscript with marked changes”.

Are the results clearly presented?

Can be improved

Accordingly, the results which should be supported conclusion have been added, for the detailed revisions, please refer to the document named “Revised manuscript with marked changes”.

Are the conclusions supported by the results?

Can be improved

Also the results have been revised, for the detailed revisions, please refer to the document named “Revised manuscript with marked changes”.

  1. Point-by-point response to Comments and Suggestions for Authors

Comments 1: Why 25g? Does this depend on the size of the reactor? This should be really clarified.

Response 1: Many thanks for this question!

   From SEM results, particle size distribution, the average widths of NMO-20, NMO-25, and NMO-30 are 240, 207, and 280 nm, respectively, with relatively wide particle size distribution. Moreover, NMO-25 exhibits the smallest particle size.

When the content of molten salt is low, although molten salt is in a liquid phase state, due to the insufficient amount of liquid phase, particles can only move for a short distance, making it difficult for particles to engulf each other, and large particles are difficult to form, resulting in smaller particle sizes. With the continuous increase of molten salt content, the liquid phase becomes more and more abundant, making it easier for particles to move over long distances. The probability of large particles swallowing small particles increases. The growth of particles in molten salt follows the Ostwald Ripening mechanism, where small grains dissolve and then precipitate on the surface of large grains, ultimately leading to the growth of large grains and the continuous increase of particle size. However, as the amount of liquid phase continues to increase, it will lead to the generation of too much liquid phase, increasing the diffusion distance and hindering grain growth.

So NMO-25 exhibits the smallest particle size, after lithiation and reduction process, LMO-25 and R-LMO-25 also show the smallest particle size. Particle size reduction can reduce the diffusion distance of lithium ion, which resulting in better electrochemical performance.

Therefore, 25 g is considered as the optimum molten salt amount.

Comments 2: The presentation of XRD results is poor. For example: fig 8. Baseline correction should be applied. Peaks should be identified.

Response 2: Thanks a lot for pointing this out!

According to the reviewer’s suggestion, baselines of XRD patterns in Figure 8 have been subtracted. Peaks in Figure 3 and Figure 8 have been identified. However, peaks in Figure 1 and Figure 6 have not been identified because the XRD patterns of Na0.44MnO2 synthesized at different temperatures and with different molten salt contents both exhibit two phases, including the orthogonal phase Na4Mn9O18 and Na0.91MnO2.

Accordingly, Figure 3 (Lines 152-154) and Figure 8 (Lines 240-242) in the manuscript were revised. For the detailed revisions, please refer to the document named “Revised manuscript with marked changes”.

Comments 3: Fig 5 and 10: it is better to show how the Li2MnO3 has been activated during first 10 cycles. This is because the initial discharge at 4.8V to about 4V is important. It will confirm that the Li2MnO3 is really reactive or you simply has a capacitor effect.

Response 3: Thanks a lot for your comment!

We compared the initial voltage-capacity curves and the corresponding differential capacity plots of Li2MnO3 and reduced Li2MnO3 in our previous work. The charge profile of reduced Li2MnO3 in Figure R1a shows two features. A remarkable feature is that, although Li2MnO3 is still the major constituent of R-Li2MnO3, its charge profile in the first cycle does not show the characteristic long voltage plateau at around 4.7 V. The disappearance of this voltage plateau should be the result of structure defects, including the stacking faults and oxygen vacancies, inside the crystalline of reduced Li2MnO3. For stoichiometric Li2MnO3, the characteristic voltage plateau in the first charge process was generally considered to be an oxygen extraction process (Li2MnO3→ MnO2 +2Li++O+e) that electrochemically activated this Mn4+-containing material. However, XRD Rietveld analysis shows a large amount of oxygen vacancies are produced by low-temperature reduction reaction and that a large amount of oxygen in the material has been extracted before the material is charged as the cathode of LIB. It might also be possible that these disordered structures can charge−discharge without being electrochemically activated in the first charge process. The other feature of the charge profile of reduced Li2MnO3 is that, compared with pristine Li2MnO3, two new voltage plateaus locating at 3.4 and 4.1 V appears in the first charge profile. The voltage plateau at 3.4 V should correspond to the Li-ion deinsertion from the orthorhombic LiMnO2 in the material. The voltage plateau at 4.1 V is not apparent but can be visible. Since no spinel phase LiMn2O4 was detected inside reduced Li2MnO3, this voltage plateau could be due to the structure transformations of LiMnO2 into a spinel-like structure, upon Li+ removal, in the first charge process.38 In Figure R1c, the voltage plateau locating at 3.4 V in the first charge falls to 3.2 V in the second charge profile, which is a characteristic for orthorhombic LiMnO2 as a cathode material for LIB. After slow and persistent decline, this voltage plateau moves to and stabilizes at ∼3.1 V in the 30th charge−discharge cycle.

Figure R1 The typical voltage-capacity curves (a) and the corresponding differential capacity plots (b) of pristine Li2MnO3 and reduced Li2MnO3 in the first charge-discharge cycle; the voltage-capacity curves of reduced Li2MnO3 in the initial 30 charge-discharge cycles.

Reviewer 2 Report

Comments and Suggestions for Authors

Following comments to the authors to improve the manuscript quality for further decision:

1.     Consider adding specific details about the molten salt used in the synthesis process to make the title more informative.

2.     For the abstract

a.      Specify the molten salt used in the synthesis method.

b.     Clarify the significance of Na0.44MnO2 as a precursor and its role in the synthesis of Li2MnO3 nanobelts.

c.      Provide a brief overview of the key findings and contributions of the study.

  1. Add more specific keywords related to the synthesis method and battery performance.
  2. The introduction section should be revised
    1. Clearly state the objective of the study and the specific problem addressed.
    2. Provide a concise literature review that highlights the current state of research on Li2MnO3 and its significance in lithium-ion batteries.
  1. In the experimental section need to add and address the below mentioned details to improve.
    1. Include more details about the molten salt used, such as its composition and any specific considerations in its selection.
    2. Provide information on the equipment and conditions used for XRD and SEM characterizations.
    3. Clarify the choice of LiNO3 and LiCl in the synthesis of Li2MnO3 nanobelts.
  1. Results and Discussion:
    • Discuss the significance of the observed XRD peaks and SEM morphologies with the electrochemical performance.
    • Present and interpret the XRD patterns and SEM images to support the conclusions.
    • Elaborate on the reasons for selecting specific reaction temperatures and molten salt amounts.
  2. In figures,
    • Ensure that the figures are labeled clearly (should be consistent in the whole manuscript) and consistently.
    • Figures size and resolution must be revised.
  3. Conclusion:
    1. Summarize the main findings and their implications for the synthesis of Li2MnO3 nanobelts.
    2. Discuss potential future directions for research in this area.

9.     Proofread the document for grammatical and typographical errors.

Example :  

        In line 94 typo error should be corrected.

        In lines135-137 authors mentioned, " Furthermore, the wide and weak diffraction peaks between 20 ~ 25° correspond to the (020), (-111), and (021) crystal planes of Li2MnO3, which are related to the superstructure formed by the ordered arrangement of Li/Mn in the transition metal layer."

Comments on the Quality of English Language

Proofread the document for grammatical and typographical errors.

Example :  

        In line 94 typo error should be corrected.

        In lines135-137 authors mentioned, " Furthermore, the wide and weak diffraction peaks between 20 ~ 25° correspond to the (020), (-111), and (021) crystal planes of Li2MnO3, which are related to the superstructure formed by the ordered arrangement of Li/Mn in the transition metal layer."

Author Response

  1. Summary

2. Questions for General Evaluation

Reviewer’s Evaluation

Response and Revisions

Does the introduction provide sufficient background and include all relevant references?

Can be improved

We have revised introduction, for the detailed revisions, please refer to the document named “Revised manuscript with marked changes”.

Are all the cited references relevant to the research?

Can be improved

The references relevant to the research have been added in the manuscript, for the detailed revisions, please refer to the document named “Revised manuscript with marked changes”.

Is the research design appropriate?

Yes

Are the methods adequately described?

Can be improved

We have modified the experimental section, for the detailed revisions, please refer to the document named “Revised manuscript with marked changes”.

Are the results clearly presented?

Can be improved

Accordingly, the results which should be supported conclusion have been added, for the detailed revisions, please refer to the document named “Revised manuscript with marked changes”.

Are the conclusions supported by the results?

Can be improved

Also the results have been revised, for the detailed revisions, please refer to the document named “Revised manuscript with marked changes”.

  1. Point-by-point response to Comments and Suggestions for Authors

Comments 1: Consider adding specific details about the molten salt used in the synthesis process to make the title more informative.

Response: Many thanks for this advice!

    The molten salts used in the synthesis process were added in the material synthesis process, which can be shown in Page 2, Lines 88-89 and Lines 94-95.

All changes/revisions/additions were highlighted in RED point-by-point with the cancelled parties labeled by Dbl Strikethrough in one copy of manuscript. (Named as “Revised manuscript with marked changes”)

For the detailed revision, please refer to document named as “Revised manuscript with marked changes”.

Comments 2: For the abstract

  1. Specify the molten salt used in the synthesis method.
  2. Clarify the significance of Na44MnO2 as a precursor and its role in the synthesis of Li2MnO3 nanobelts.
  3. Provide a brief overview of the key findings and contributions of the study.

Response: Thanks a lot for your suggestions!

The molten salt used in the synthesis method has been specified, the significance of Na0.44MnO2 as a precursor template and manganese source and a brief overview of the key findings and contributions of the study have also been provided in the abstract, which can be shown in Page 1, Lines10-12 and Lines 24-25.

For the detailed revision, please refer to document named as “Revised manuscript with marked changes”.

Comments 3: Add more specific keywords related to the synthesis method and battery performance.

Response: Thanks a lot for valuable advice!

   The keywords in our manuscript have been modified as “Na0.44MnO2; Li2MnO3; nanobelts; Molten-salt method; Rate performance; Cathode material”. For the detailed revision, please refer to document named as “Revised manuscript with marked changes”, which can be found in Page 1, Lines 26-27.

Comments 4: The introduction section should be revised.

  1. Clearly state the objective of the study and the specific problem addressed.
  2. Provide a concise literature review that highlights the current state of research on Li2MnO3 and its significance in lithium-ion batteries.

Response: Thank you for pointing these out!

We have modified the introduction section to state the objective of the study and the specific problem addressed and provide a concise literature review that highlights the current state of research on Li2MnO3 and its significance in lithium-ion batteries.

For the detailed revision, please refer to document named as “Revised manuscript with marked changes”, which can be found in Page 2, Lines 60-69 and Lines 72-79.

Comments 5: In the experimental section need to add and address the below mentioned details to improve.

  1. Include more details about the molten salt used, such as its composition and any specific considerations in its selection.
  2. Provide information on the equipment and conditions used for XRD and SEM characterizations.
  3. Clarify the choice of LiNO3 and LiCl in the synthesis of Li2MnO3

Response: Many thanks for your suggestions!

The molten salts used have been added, and in the process of Na0.44MnO2 nanobelts synthesis, NaCl was used as molten salt according to previous work. In the process, the mixture of LiNO3 and LiCl was used as molten salt. At the same time, LiNO3 was used as the oxidizing reagent. For the detailed revision, please refer to document named as “Revised manuscript with marked changes”, which can be shown in Page 2, Lines 88-89, Lines 94-95 and Page 3, Lines 99-101.

Information on the equipment and conditions used for XRD and SEM characterizations has been revised in the materials characterization. For the detailed revision, please refer to document named as “Revised manuscript with marked changes”, which can be shown in Page 3, Lines 107-109

In the synthesis of Li2MnO3 nanobelts, numerous literatures have proven that Na0.44MnO2 is converted into lithium manganese oxide in LiNO3-LiCl eutectic molten salt. When the temperature was 450 ℃, the XRD pattern confirmed that the main phase agreed with Li0.44MnO2, and the second phase was Li2MnO3. However, the XRD pattern of the product performed at 500 ℃ was pure-phase Li2MnO3. So, we choice the mixture of LiNO3-LiCl, and the synthesis temperature is 500℃.

Additionally, in this process, Li2MnO3 involved the ion-exchange between Li+ in molten LiNO3 and LiCl salts and Na+ in Na0.44MnO2 and the oxidation process by LiNO3.

Comments 6: Results and Discussion:

  1. Discuss the significance of the observed XRD peaks and SEM morphologies with the electrochemical performance.
  2. Present and interpret the XRD patterns and SEM images to support the conclusions.
  3. Elaborate on the reasons for selecting specific reaction temperatures and molten salt amounts.

Response: Many thanks for these valuable suggestions!

Accordingly, we have added the significance of the observed XRD peaks and SEM morphologies with the electrochemical performance. For the detailed revision, please refer to document named as “Revised manuscript with marked changes”, which can be found in Page 7, Lines 206-210, Lines 214-215, and Lines 217-218. For the effects of molten salts contents, there is little effect on the crystallinity of all the samples with different amounts of molten salt. And when NaCl is 25g, the particle sizes of Na0.44MnO2, Li2MnO3 and reduced Li2MnO3 are the smallest.

   The XRD and SEM results have been added in conclusion to support the conclusions. For the detailed revision, please refer to document named as “Revised manuscript with marked changes”, which can be found in Page 12, Lines 341-344.

   Na0.44MnO2 nanobelts were synthesized by molten-salt method according to the published procedures [1], 1 mmol of MnCO3 an 0.44 mmol of Na2CO3 were mixed with 5 g NaCl, ground homogeneously in a mortar for 30 min. The mixture was then placed into an alumina crucible, and annealed at 850 ℃ for 5 h in a crucible furnace. And the results showed that Na0.44MnO2 with morphology of nanobelts with diameters ranging from 100 nm to a few hundred nanometers and length up to tens of micrometers. In order to study the growth mechanism of Na0.44MnO2 nanobelts precursor, together with the relationship between the synthesis conditions of precursor and electrochemical performances of reduced Li2MnO3, so we choose the temperature ranging from 800~900 ℃, the molten salts contents of 5g, 10g, 15g, 20g, 25g, 30g, 40g and 50g. From our SEM images and the average widths of 5g, 10g, 15g, 20g, 25g, 30g, 40g and 50g, as the content of molten salt increases, the particle size first increases and then decreases.

[1] Zhang, X. K.; Tang, S. L.; Du, Y. W. Controlled Synthesis of Single-Crystalline Li0.44MnO2 and Li2MnO3 Nanoribbons Mater. Res. Bull. 2012, 47, 1636-1640.

Comments 7: In figures,

  1. Ensure that the figures are labeled clearly (should be consistent in the whole manuscript) and consistently.
  2. Figures size and resolution must be revised.

Response: Many thanks for this advice!

   All the figures in manuscript were labeled clearly and consistently, and the figures size and resolution were revised carefully.

All changes/revisions/additions were highlighted in RED point-by-point with the cancelled parties labeled by Dbl Strikethrough in one copy of manuscript. (Named as “Revised manuscript with marked changes”)

For the detailed revision, please refer to document named as “Revised manuscript with marked changes”

Comments 8: Conclusion:

  1. Summarize the main findings and their implications for the synthesis of Li2MnO3
  2. Discuss potential future directions for research in this area.

Response: Thanks a lot for these suggestions!

    According to the suggestions, the sentences “When the reaction temperature was 850 ℃, Na0.44MnO2 precursor was uniformly distributed with smaller particle size of 240 nm. The effect of molten salt contents on the crystallinity was not significant, when the molten salt content was adjusted to 25 g, Na0.44MnO2 precursor showed the smallest particle size of 204 nm.” and “The growth mechanism has guiding significance for the synthesis of materials with special morphologies by molten salt method. These findings are expected to provide a feasible approach for preparing special morphology cathodes for LIBs.” have been added in conclusion.

For the detailed revision, please refer to document named as “Revised manuscript with marked changes”, which can be shown in Page 12 , Lines 321-344 and Lines 349-352.

Comments 9:  Proofread the document for grammatical and typographical errors.

Example:  

  • In line 94 typo error should be corrected.
  • In lines135-137 authors mentioned, " Furthermore, the wide and weak diffraction peaks between 20 ~ 25° correspond to the (020), (-111), and (021) crystal planes of Li2MnO3, which are related to the superstructure formed by the ordered arrangement of Li/Mn in the transition metal layer."

Response: Thanks a lot for pointing this out!

The document for grammatical and typographical errors, such as Line 94 and Lines 135-137 have been revised, which can be shown in Line 100 and Lines 146-148.

For the detailed revision, please refer to document named as “Revised manuscript with marked changes”.

Comments on the Quality of English Language

Proofread the document for grammatical and typographical errors.

Example:  

  • In line 94 typo error should be corrected.
  • In lines135-137 authors mentioned, " Furthermore, the wide and weak diffraction peaks between 20 ~ 25° correspond to the (020), (-111), and (021) crystal planes of Li2MnO3, which are related to the superstructure formed by the ordered arrangement of Li/Mn in the transition metal layer."

Response: Thanks a lot for this comment!

Accordingly, this manuscript was further revised carefully, and several incorrect phrases, syntax, inappropriate choice of words and sentence structures were revised as shown in the revision.

For the detailed revision, please refer to document named as “Revised manuscript with marked changes”.

Round 2

Reviewer 1 Report

Comments and Suggestions for Authors

The authors did a lot of work in responding to the reviewer request.

The experimental section should be more detailed. Also fig R1 should be added, maybe to a SI.

Author Response

  1. Summary Thank you very much for taking the time to review this manuscript again. Those comments are all valuable and very helpful for revising and improving our paper, as well as important guides for our research. We have studied the comments carefully and have made corrections. Revised portions are marked in red on the paper.
  2. Questions for General Evaluation Reviewer’s Evaluation Response and Revisions Does the introduction provide sufficient background and include all relevant references? Yes Are all the cited references relevant to the research? Yes Is the research design appropriate? Yes Are the methods adequately described? Yes Are the results clearly presented? Yes Are the conclusions supported by the results? Can be improved Also the results have been revised, for the detailed revisions, please refer to the document named “Revised manuscript with marked changes”, Which can be found in Page 12, Lines 350-353.
  3.  Point-by-point response to Comments and Suggestions for Authors Comment 1: The experimental section should be more detailed. Also fig R1 should be added, maybe to a SI. Response 3: Thanks a lot for your comment! The experimental section has been added, the sentences “The prepared Li2MnO3 nanobelts, stearic acid, and an appropriate amount of absolute ethanol were ground for a few minutes to get a uniform solid-liquid rheological material. Then obtained rheological material was heated at 340 ℃ in an Ar atmosphere for 8 hours. After washing with several times, the product was obtained.” have been added in our manuscript. For the detailed revisions, please refer to the document named “Revised manuscript with marked changes”, which can be found in Page 3, Lines 103-108. Images and the corresponding comments similar to Figure R1 have already appeared in our previous works. In order to avoid repeated use of images and conclusions, we have not added Figure R1 to SI. In addition, we consider that this part of work mainly discusses the influence of Na0.44MnO2 precursor synthesis conditions on the electrochemical performance of reduced Li2MnO3, so the focus is not on the difference of electrochemical behavior and activation process between pristine and low-temperature reduction Li2MnO3. I am sorry for this. The corresponding sentence “Additionally, the curves of three reduced samples are quite different from pristine Li2MnO3, which have been confirmed in our previous works27-29.” has been added, which can be shown in Page 6, Lines 201-203.

Reviewer 2 Report

Comments and Suggestions for Authors

Authors improved manuscript quality and revised (highlighted revised information).